# Learning to Hunt on the Go: Dietary Changes During Development of Rhinolophid Bats

**DOI:** 10.3390/ani14223303

**Published:** 2024-11-16

**Authors:** Miren Aldasoro, Nerea Vallejo, Lander Olasagasti, Oihane Diaz de Cerio, Joxerra Aihartza

**Affiliations:** Department of Zoology and Animal Cell Biology, University of the Basque Country UPV/EHU, Sarriena z/g, 48940 Leioa, Basque Country, Spain; nerea.vallejo@ehu.eus (N.V.); lander.olasagasti@ehu.eus (L.O.); oihane.diazdecerio@ehu.eus (O.D.d.C.); joxerra.aihartza@ehu.eus (J.A.)

**Keywords:** development, diet, horseshoe bats, metabarcoding, traits, trophic niche partitioning

## Abstract

The diet of mammals changes as they grow. This study explores how the diets of three horseshoe bat species change as they grow from juveniles to adults. By analysing the DNA in their droppings, we compared the diets of juveniles and adults at two levels: the types of prey that they eat and the characteristics of their prey (size, hardness, and flying speed). We found differences between the two age groups in two out of the three species. A common pattern emerged, where younger bats tend to eat smaller, easier-to-hunt prey, suggesting that hunting skills and other factors like learning may develop differently across species. These findings have implications for conservation efforts of these bat species, particularly in recognising the distinct dietary needs of juveniles for their survival and successful development.

## 1. Introduction

As animals grow, they experience changes in physiology and competitive pressure, resulting in different dietary niches at various life stages (e.g., [1,2]). Age-related changes in diet have implications for energy budgets, competitive interactions, and predator–prey dynamics, affecting individuals and groups at different life stages. Body size is a central parameter in determining prey selection, and when animal size increases, ontogenetic dietary shifts are usual [3].

Ontogenic dietary changes in bats have been studied very little, and most research has focused on vespertilionid bats (e.g., [4,5]), with only limited data available on rhinolophid bats [6]. Studying these diet differences can provide valuable information on bat development, behaviour, ecosystem dynamics, and species resilience. Juvenile bats may require different nutrients to support rapid growth and development, and studying these needs can help us to understand how diet supports the life cycle stages of bats. Moreover, juvenile bats might have less-developed foraging skills, possibly leading to a diet that is less diverse or lower in quality, and they may forage closer to the roost initially, consuming easier or more available food sources. Thus, observing diet differences could highlight how feeding skills and preferences evolve with age in bats, and how bats transition to independence and develop their hunting and navigation skills.

Along with their body size and flight abilities, echolocation plays a crucial role in bats’ hunting skills. Echolocation significantly influences their foraging behaviour, determining where they hunt and their ability to detect prey based on size or flight patterns (e.g., [7,8,9,10]). During the period between their first flight and weaning, young bats must go through a learning process to perfect their navigation, flight, and hunting skills [11,12]. This learning includes improving their ability to catch prey and memorising the locations of distant food resources [13].

Optimal foraging theory [14] is valuable for understanding feeding behaviour and predicting dietary changes. According to this theory, hunting is only worthwhile if the energy gained exceeds the energy spent. Predator–prey interactions depend on prey profitability and the effectiveness of defensive strategies [15,16]. Factors like prey size or evasiveness affect profitability, while prey availability and predators’ niche flexibility shape diet [17]. Traits such as size, sex, and individual predator specialisation can influence feeding habits. Analysing these functional traits provides new insights into foraging ecology and behaviour in complex predator–prey systems [6].

Horseshoe bats emit echolocation calls with long constant frequency (CF) components [18,19]. These characteristic calls may allow them to forage selectively in the wild, distinguishing between different types of prey based on information about, for example, the speed of insect wing beats contained in the echoes [20]. However, using echolocation to hunt requires a learning process, which also goes hand in hand—and likely correlates—with vocalisation development. Different studies have shown that juvenile echolocation calls’ resting frequency (RF) was significantly lower than that of adults [21,22,23], affecting prey identification capability. Infant horseshoe bats develop Doppler shift compensation (DSC) [24], with limited capacity emerging around 30 days and reaching adult-like levels by 40–45 days in *Rhinolophus ferrumequinum*. Since juveniles begin feeding at 28–30 days [25], they may initially rely on incomplete DSC, potentially compromising their echolocation abilities.

In the Basque Country (Northern Iberian Peninsula), *Rhinolophus euryale*, *R. ferrumequinum*, and *R. hipposideros* cohabit, each specialising in hunting in cluttered environments to varying degrees [26,27]. *R. ferrumequinum* is the largest species, emitting echolocation calls at 79–84 kHz, while *R. hipposideros*, the smallest, calls at 106–116 kHz [26]. *R. euryale*, the intermediate-sized species, emits 102–109 kHz calls [27].

*Rhinolophus euryale* has the most specialised diet, focusing on moths of various sizes, with juveniles reported as targeting slower, smaller moths, allegedly due to psychomotor learning. This process involves the development of motor skills and coordination through practice and experience [6]. In contrast, *R. ferrumequinum* has a diverse diet, including large prey like coleopterans, which, despite high energy costs, provide substantial energy returns [19,28,29]. Finally, the diet studies hitherto showed that *R. hipposideros* primarily hunts small prey, such as dipterans, lepidopterans, and hemipterans, indicating a narrower scope of prey size. [30,31,32]

In this study, we compare the diets of juveniles and adults across the three species, testing the following hypotheses: (1) there will be significant dietary differences between juveniles and adults within each species, which will diminish as juveniles mature; (2) since foraging skills are tied to learning orientation, flying, and prey capture [11,12], juveniles will have a less specialised and more opportunistic diet, focusing on prey that is easier to catch and handle; (3) species with higher trophic specialisation will show more significant diet differences between juveniles and adults than those with lower specialisation. Additionally, we aim to identify the core dietary needs of juveniles in these three horseshoe bat species, which are crucial for their survival during the first winter and essential for conservation efforts.

## 2. Materials and Methods

### 2.1. Study Area

The study was conducted in three bat colonies, each with over 100 individuals, minimising sampling impact. The roosts, all within 100 km of each other in the Basque Country (Northern Iberian Peninsula), include *Rhinolophus ferrumequinum* in a church and *R. euryale* in a limestone cave, both in Enkarterriak, a hilly region (200–855 m a.s.l.) with a mosaic of meadows, pastures, and dense hedgerows. The *R. hipposideros* colony roosts in a church in Matxinbenta—Gipuzkoa—surrounded by deciduous and conifer forests, meadows, and pastures. The study area experiences a temperate Atlantic oceanic climate.

### 2.2. Sample Collection

Sampling was conducted on a single night in each roost to minimise disturbances to the colonies during the breeding season: early August 2021 for *R. ferrumequinum*, early August 2022 for *R. hipposideros*, and mid-August 2022 for *R. euryale*. Bats were captured using a 2 × 2 m hand-made harp trap [33] set at roost entrances from 00:30 h. onwards, targeting bats returning from foraging. We placed each captured bat in a clean cloth bag until it defecated (40–90 min). Faecal material was stored in 1.5 mL Eppendorf tubes with drying paper and frozen at −80 °C within six hours. Individual bats were considered as sample units [34].

### 2.3. Age and Development Parameters

Individuals were sexed (SEX), weighed (WG), and measured for forearm length (FL) using a scale to the nearest 0.1 g and a calliper to the nearest 0.1 mm. Age class (juveniles or adults) was determined by fur colour (light grey for juveniles) and the presence or absence of the epiphyseal gap (EG). Juveniles’ epiphyses were photographed with the wing spread above a strong torch, and the fifth phalanx was measured. Resting frequencies (RFs) of their echolocation calls were recorded in hand with a u256 USB Ultrasound Microphone (Pettersson Elektronik, Upsala, Sweden) for later analysis. The captured bats were then released into their roosts.

As indicators of the development level of juveniles, we relied on the EG of their wing joints and the constant frequency component of their echolocation calls (RF). The EG was measured on the photographs (in millimetres) (Adobe Photoshop 2019, Adobe Inc., San José, CA, USA) by summing the two observable—translucent—cartilage bands corresponding to the proximal and distal epiphyseal or growth plates (based on [35]) (Appendix A). The RF was ascribed by analysing power spectrograms (FFT size: 2048; window type: Hanning) and identifying the dominant frequency of the calls using BatSound (v 4.03) [36]. We explored the correlations between the development parameters of each species and the potential sexual dimorphism (Appendix A).

### 2.4. DNA Extraction, PCR Amplification, Library Preparation, and Sequencing

DNA was extracted from frozen individual faecal samples (up to 280 mg) using the Kingfisher extraction tool (Thermo Fisher Scientific, Waltham, MA, USA). Two mini-COI segments (178 and 180 bp) of mitochondrial DNA were amplified using FWH1 (fwhF1/fwhF2) [37] and ANML (CO1490/CO1-CFMRa [38] primers. Amplifications were performed with the QIAGEN Multiplex PCR Kit (Qiagen Iberia, S.L. Barcelona, Spain) in 25 μL reactions, and PCR success was confirmed by agarose gel electrophoresis. A second PCR attached unique tags and Illumina adapters to each amplicon. Finally, 137 samples were pooled and sequenced using Illumina MiSeq (v2 PE, 500 cycles, 15 M reads). DNA library construction and sequencing processes were carried out at the Genomics and Proteomics General Service (SGIker) of the University of the Basque Country.

### 2.5. Bioinformatic Procedure for Operational Taxonomic Units (OTU) Analyses

Separation by primers, quality control, sequence pre-processing, and collapsing identical sequences into single ones were performed using CUTADAPT [39] and USEARCH [40]. We clustered sequences into operational taxonomic units (OTUs) by VSEARCH [41] at a 97% similarity threshold [42] using the “–cluster_size” command. Subsequently, we cleaned up chimaera OTUs with VSEARCH’s “-uchime3_denovo” command.

The taxonomic assignment of each OTU was performed by comparing the representative sequence against reference sequences in the Barcode of Life Database (BOLD; www.boldsystems.org) and GenBank (www.ncbi.nlm.nih.gov), both accessed on 8 November 2022. The “Boldigger-cline” command [43] was used for BoldSystems and the “-blastn command” [44] for GenBank. For the latter, only hits with a pairwise identity above 98% and e-values below 1 × 10^−20^ were accepted [45,46] to ensure that the match did not occur by chance. The databases of arthropods for Spain, France, and Portugal were downloaded from GBIF (www.gbif.org; acceded on 3 November 2022) to verify that the identified species encompassed our study area. In OTUs with multiple potential assignments, identifications were carefully reviewed and refined manually to assign the most accurate taxa to each OTU based on pairwise identity and species distribution.

### 2.6. Statistical and Data Analysis

#### 2.6.1. Description of Dietary Metrics

Only the OTUs classified as potential prey were used for the diet analysis, with data converted to weighted percentages of occurrences (wPOOs), a metric based on presence/absence data that provides a good proxy of consumption [47,48]. Core diets were analysed by selecting the species with a frequency of occurrence (FOO) above 5% [29] in each colony.

#### 2.6.2. Effect of Bat Species and Age on the Taxonomic Diet Composition

We explored the differences between the diets of adults and juveniles through PERMANOVA [49] using function adonis. We performed NMDS (non-metric multi-dimensional scaling) with the metaMDS function to visualise species composition differences among samples. We compared the species’ intraspecific resource partitioning using Pianka’s [50] measure of niche overlap with the Pianka function at the species level. We calculated the Bray–Curtis dissimilarities between all samples using function vegdist [51].

#### 2.6.3. Functional Traits Analysis for Predator and Prey

We focused the trait analyses on the bat species’ core diets. A total of 93 out of the 119 species constituting the core diets (Appendix A) were observed in situ in the Entomology Collection of the Museo Nacional de Ciencias Naturales—CSIC (MNCN-CSIC, Madrid). They were photographed with a millimetre scale for measuring key traits: volume proxy excluding wings (VOL = length × width) and wing loading (WL = weight/forewing area) as an indicator of flying speed (following [6,52]). Hardness values (HRD) were calculated using Freeman and Lemen’s [53] equation and categorised into four levels. When prey items were unavailable, trait data were estimated at the genus or family level when possible (Appendix A).

We studied the functional relationships between bat characteristics and prey traits using RLQ analysis combined with fourth-corner analysis [54] using the ade4 package for R version 4.2.0 
[55]. This method helps to clarify horseshoe bats’ complex ecology and diverse prey [6]. Traits were log-transformed (X’ = log (X + 1)) for comparability. The RLQ analysis involved constructing three starting matrices, following Arrizabalaga-Escudero et al. [6] (Appendix A). *p*-values were adjusted by Benjamini and Hochberg [56] correction method. The analysis was conducted separately for each bat species, comparing adult and juvenile diets.

## 3. Results

### 3.1. Diet Composition at the Taxonomic Level

We captured and sampled a total of 47 *Rhinolophus ferrumequinum* (22 juveniles, 25 adults), 44 *R. hipposideros* (23 juveniles, 21 adults), and 45 *R. euryale* (26 juveniles, 19 adults). We successfully extracted DNA from all *R. ferrumequinum* and *R. hipposideros* individuals, and 42 *R. euryale* (23 juveniles, 19 adults). After the bioinformatic analysis, we obtained 7271 OTUs. Even though most of them could not be identified—or were identified at higher taxonomic levels—we ascribed 1681 of them to the species or genus levels. After removing the contamination and collapsing repeated identifications, we identified 271 putative prey species (sp) belonging to 75 families of 10 different orders: Lepidoptera (149 sp), Diptera (72 sp), Hemiptera (16 sp), Coleoptera (12 sp), Neuroptera (12 sp), Trichoptera (4 sp), Araneae (3 sp), Ephemeroptera (1 sp), Blattodea (1 sp), and Orthoptera (1 sp) (Appendix A).

In *Rhinolophus euryale*, adults were highly specialised as their diet was almost exclusively based on Lepidoptera (wPOO = 96.23%) (Figure 1). Juveniles, instead, showed a less specialised diet with a greater spectrum of prey orders: lepidopterans had a lower wPOO value (68.65%). In contrast, dipterans and neuropterans increased their importance in juveniles’ diets (20.62% and 7.68%, respectively).

The main prey of adult *R. hipposideros* were Lepidoptera (wPOO = 47.12%), Diptera (38.69%), and Hemiptera (13.27%) (Figure 1). In juveniles, though, the diet was primarily composed of dipterans (52.90%), followed by fewer lepidopterans and hemipterans (30.75% and 4.11%, respectively). In comparison, the importance of neuropterans and trichopterans increased (6.19% and 3.99%, respectively).

In *R. ferrumequinum*, adults showed a balanced diet, mainly composed of Lepidoptera (wPOO = 38.64%), Coleoptera (26.38%), and Diptera (26.11%) (Figure 1). The juveniles’ diet was also based on these orders, with only slight differences in their values: Lepidoptera gained some significance (46.59%) and Diptera decreased (19.66%), while Coleoptera remained the same (26.78%).

When comparing the species-level taxonomic composition of the three bat species diets, the PERMANOVA analyses showed significant differences between the diets of juveniles and adults in *R. euryale* (F = 6.2962, *p* < 0.001) and *R. hipposideros* (F = 2.8792, *p* < 0.001) but not so strong differences in *R. ferrumequinum* (F = 1.4858, *p* = 0.090) (Figure 1).

### 3.2. Niche Overlap Analysis

The intraspecific niche overlap or similarity percentage among juveniles and adults diets, based on wPOO at the prey species level, showed the most distinct niches in *R. euryale* (Pianka’s index of 16.70%). *R. hipposideros* presented higher similarity, with an overlap value of 61.90%, while *R. ferrumequinum* only revealed slight differences between age classes (Pianka’s index of 81.93%). The NMDS analysis shows the same pattern, with the juvenile and adult diet compositions overlapping the least in *R. euryale* (Figure 2a), intermediately in *R. hipposideros* (Figure 2c), and largely in *R. ferrumequinum* (Figure 2b).

### 3.3. Predator–Prey Trait Analysis

#### 3.3.1. *Rhinolophus euryale*

When we analysed the *R. euryale* diet at the prey trait level, RLQ analysis clearly distinguished two blocks of bats, with the adult individuals on the left side and the juveniles on the opposite (Figure 3c). The first two axes of the RLQ ordination accounted for 99.06% (axis Q1/R1, horizontal; Figure 3a) and 0.82% (axis Q2/R2, vertical; Figure 3b) of the total percentage or relation (coinertia) between bats’ traits and moths’ traits, respectively. In this case, axis 1 explained almost all of the relation. The length of the arrows is proportional to the variation explained by a given trait. The fourth-corner analysis related AGE, WG, FL, and RF positively with hardness and volume and negatively with EG. The combination of the RLQ and the fourth-corner test confirmed the above-mentioned functional diet change during the bats’ development: all bat traits except sex were associated with axis Q1; hardness and volume were associated with axis R1 (Appendix A).

The relative positions of arrows hardness and volume on the right side of the horizontal axis (Figure 3b) and the arrows representing bats’ traits such as age or FL (Figure 3a) indicate a general transition from smaller and smoother prey items to harder and bigger ones as the bats grow up; or, taxonomically, the shift from the prey hunted by juveniles (mainly dipterans and neuropterans such as *Hybos culiciformis*, *Neolimonia dumetorum*, *Rhipidia maculata*, or *Hemerobius gilvus*) to the ones hunted by adults (noctuids *Amphipyra pyramidea* and *Mythimna unipuncta*, and erebid *Aedia leucomelas)* (Appendix A).

#### 3.3.2. *Rhinolophus hipposideros*

Concerning *R. hipposideros*, the measured bat traits revealed sexual dimorphism in this species (see Appendix A), with females emitting at higher echolocation frequencies and having longer forearms than males. Consistently, the RLQ analysis showed a double grouping (Figure 4c): first, females grouped in the upper quadrants and males grouped in the lower ones, and, second, adults grouped on the right side and juveniles on the left. The first two axes account for 96.35% (axis Q1/R1, horizontal; Figure 4a) and 3.47% (axis Q2/R2, vertical; Figure 4b) of the total coinertia. The fourth-corner analysis correlated volume positively with age and RF. At the same time, it was negatively related to sex (males consuming prey with smaller volume). In addition, hardness was positively correlated to RF (Figure 4d). The combination of the RLQ and the fourth-corner test could confirm the functional diet change of the developing bats (*p* < 0.05): sex, age, WG, RF, and EG were associated with axis Q1, while volume was associated with axis R1 (Appendix A). Note that some adults (seven males and four females) were excluded from the analysis due to the lack of morphofunctional data.

The arrow of the trait epiphyseal gap (EG) points left, separating the juveniles in the left quadrants from the adults in the right quadrants (Figure 4a). The relative position of arrow sex on the upper side of the vertical axis shows that the individuals in the upper quadrants represent males, while females are in the lower quadrants. FL points down, in association with sex, since females were larger than males.

In the lower right quadrant, where the adult females are located (Figure 4a,c), we found the biggest, hardest, and fastest (corresponding to high wing loading) prey species (Figure 4b), such as the hemipterans (*Orientus ishidae*, *Adelphocoris lineolatus*), big tipulids (*Tipula fulvipennis*), and big lepidopterans such as *Mythimna unipuncta* (Appendix A). Conversely, in the lower left quadrant, juvenile females (Figure 4a) preyed upon smaller, slower (lower wing loading) lepidopterans such as *Triphosa tauteli*, *Idaea biselata*, or *Protodeltote pygarga* (Appendix A).

Whereas adult males (upper right quadrant, Figure 4a,c) hunted prey with a higher wing loading (Figure 4b)—such as small tortricids and crambids (*Celypha* sp., *Nomophila noctuella*) and dipterans (*Rhipidia maculata* and *Rhamphomyia* sp.)—juveniles ate prey with smaller wing loading, such as the trichopteran *Tinodes maculicornis* (Appendix A).

#### 3.3.3. *Rhinolophus ferrumequinum*

In the case of *R. ferrumequinum*, the RLQ analysis showed discrete clusters for juveniles and adults: adults were grouped in the right quadrant; in disparity, juveniles showed a more dispersed pattern, with the most developed individuals located around the graph centre and the less developed ones spreading to the left quadrant (Figure 5c). The first two axes account for 95.18% (axis Q1/R1, horizontal; Figure 5a) and 3.82% (axis Q2/R2, vertical; Figure 5b) of the total coinertia. The fourth-corner analysis did not show any significant association. Instead, the combination of the RLQ and the fourth-corner test only showed a weak association (*p* < 0.1) for resting frequency (RF) to axis Q1 and hardness with axis R1 (Appendix A).

The relative positions of the arrows epiphyseal gap (EG) in the upper left quadrant, opposed to the arrow resting frequency (RF) in the right quadrant, separate the individuals by age: the youngest and least developed individuals are in the left quadrants, while the adults appear in the right ones (Figure 5a). These arrows are related to the prey traits wing loading and hardness (Figure 5b), indicating that there was a tendency for these individuals to hunt increasingly faster and harder prey as they developed, linked to the consumption of big coleopterans such as *Prionus coriarius*, *Arhopalus ferus*, and *Arhopalus rusticus* (Appendix A). Moreover, there is a clear separation by sex in the vertical axis, where females mainly occupy the upper quadrant while the males are on the bottom. This grouping suggests that males consume bigger but slower (lower wing loading) prey, like those mentioned above, than females.

## 4. Discussion

Our study presents unique and novel findings on the dietary differences between adult and juvenile bats in *Rhinolophus euryale* and *R. hipposideros*, with *R. ferrumequinum* showing no such clear distinction. These results partially meet our initial hypothesis. At the trait level, we found correlations between bats’ developmental indicators (body size, echolocation frequency, epiphyseal gap) and the characteristics of their most consumed prey, further highlighting the distinct dietary patterns in each species.

Our study, conducted on a single sampling day to minimise disturbances to the colonies for conservation constraints, is inherently limited to the prey availability on those specific days. This limitation, while acknowledging the potential oversight of some prey species, still enables us to identify their diet differences and the prey traits involved.

Our results show that juveniles of *Rhinolophus hipposideros* and *R. euryale* species have broader diets than adults, while adults of *R. ferrumequinum* show broader diets than juveniles. When we compared the diets at the taxonomic level, we found significant intraspecific differences among juveniles and adults in *R. hipposideros* and *R. euryale*, confirming the variation reported by Arrizabalaga-Escudero et al. [6] for the latter. Moreover, in our study, *R. euryale* showed the highest dissimilarity between juveniles and adults, with the adults showing the most specialised and distinct diet. Contrarily, no significant difference was found between the diets of adults and juveniles in *R. ferrumequinum*, likely due to this species’ more generalist—less specialised—diet at the broad taxonomical level [57].

Trait-based diet studies play a crucial role in understanding the autecology of species more finely (e.g., [11,57,58]). Our dietary comparison between adult and juvenile bats contributes to this understanding and provides insights into the ecological needs and optima depending on development or sex in a more refined way (e.g., [6,15,59]).

Of the three species studied herein, differences between juveniles’ and adults’ diets at the trait level had only been previously reported in *R. euryale* perfunctorily [6]. Interestingly, our results confirm the importance of such differences but do not fully coincide with the prey traits involved in them. Thus, Arrizabalaga et al. [6] reported juvenile *R. euryale* consuming smaller, lighter moths with lower wing loading (slower flight) than adults. Our data confirmed the importance of prey size and weight in the diet shift; nevertheless, in our study, prey’s wing loading did not make any difference.

Even though both studies were conducted in the same locality and season, it is already well known that bats efficiently react to short-term changes in prey availability [60,61,62]. Thus, punctual differences in prey availability across the years—and the subsequent prey selections by adults and juveniles—could explain the different traits unveiled in our study. Nevertheless, the differences observed in both studies may also address methodological issues [63]. *R. euryale* specialises in preying on moths as an adult [64,65]. Consequently, in their molecular diet study, Arrizabalaga-Escudero et al. [6] solely relied on Zeale’s primers for sequencing, which are known for their strong bias towards Lepidoptera [28,66]. In contrast, our study combined two primer sets with a broader target spectrum and a novel technique with a higher sequencing depth, providing a more comprehensive—and therefore diverse—assessment of the *R. euryale* diet. We have shown that juveniles are not so specialised and consume a more varied diet, including other prey groups than moths. Therefore, the lack of a significant correlation between wing loading and development traits in our results may be related to the more diverse diet composition that our molecular approach unveiled.

Interestingly, juvenile *R. euryale*’s not-so-specialised and more varied diet—consuming more Diptera, Neuroptera, or Hemiptera—fits well with the diet of adult females described by Goiti et al. [67] in the same colony in a cool summer when moth availability was particularly low and the colony failed to breed. This comparison further supports that juveniles’ diet shows a suboptimal condition that they must overcome through development.

Regarding the remaining horseshoe bat species, the role of sexual dimorphism in *R. hipposideros* was outstanding. Just as adults fed on harder, larger, and faster prey (higher wing loading) than juveniles, females also preyed upon bigger, harder prey than males during the breeding season. Sexual size dimorphism has been observed in various bat species, ranging from vespertilionids [68] to horseshoe bats [69,70,71]. Moreover, even if it was not significant, we also observed a similar pattern in *R. ferrumequinum* (see the positions of males’ and females’ diets in Figure 5c).

This sexual dimorphism can be attributed to the larger size and higher energy requirements of females during the breeding season and provides a perfect example of intraspecific niche partitioning in insectivorous bats. It is known that some vespertilionid species show altitudinal segregation during the breeding season, where females use warmer roosts in more productive lowlands while males exploit higher, less productive areas, with colder roosts (e.g., [72,73,74,75]). However, this behaviour reflects habitat rather than trophic niche partitioning, the latter being directly related to the differential exploitation of one type of prey over another. Hitherto, such intraspecific trophic niche partitioning has only been described in *Tadarida teniotis* [76]. Nevertheless, a study on echolocation calls has revealed that *R. ferrumequinum* exhibits differences in call frequencies between sexes [77]. While the study primarily focused on social behaviour, these findings also indicate that the distinct use of frequencies between sexes may result in trophic niche partitioning.

Regarding the differences in diet linked to development in *R. hipposideros*, our results are consistent with those observed in *R. euryale* and reveal a general pattern where juveniles feed on smaller, slower (low wing loading), or smoother prey that are easier to catch and handle [4,5,6]. These results suggest a learning process before reaching the specialisation of adults capable of hunting and handling larger, harder, and faster (high wing loading) prey.

In the case of *R. ferrumequinum*, though, we only observed a tendency for juveniles and adults to group separately through the trait analysis. Still, that trend did not reach statistical significance. This lack of easily noticeable age effects may be due to several reasons. First, our data for this species may include a broader range of juveniles, including more developed individuals, which makes it difficult to see the differences. Second, the observed results are also likely due to adult members of this species being less selective in their diet and consuming a wider range of food.

Similarly, other studies on generalist bats found no differences between different age groups either [78]. According to the “niche variation hypothesis” [79], species with a broad ecological niche are composed of individuals with specific dietary preferences. This means that species with a broader niche exhibit higher ecological variation within their own populations [57,80,81]. Therefore, in this case, it would be more challenging to identify differences between age classes at the population level.

The varying results gathered when comparing adult and juvenile diets in horseshoe bats may have different reasons underneath, and the importance of each aspect may differ from species to species. On the one hand, the optimal foraging theory [14] foresees that adult horseshoe bats, with better hunting and flight skills, target larger, more profitable prey, avoiding smaller ones. In contrast, juveniles, less skilled and with slower flight, feed opportunistically on smaller, smoother prey, as more energy-rich prey may be out of their reach. This shift in diet likely reflects a learning process where young bats must develop foraging skills, improving their flight, orientation, and prey-capturing abilities [11,12], possibly through psychomotor learning and increased fitness [82].

Moreover, horseshoe bats’ specialised echolocation system allows them to identify and localise prey based on acoustic cues [83]. Horseshoe bats differentiate their prey by listening to their wing-beating rhythms, identified as “acoustic glints” in the echoes of the echolocating signals that bats emit [84]. This level of discrimination, though, implies that horseshoe bats must learn which wing-beating rhythms correspond to each prey species before deciding to select the most profitable prey.

Differences in foraging range and habitat also influence prey availability (e.g., [85,86,87]). A small foraging range reduces travel costs but may also be constrained by poor flight or navigational skills [88]. Juvenile horseshoe bats stay close to their maternity roost, expanding their range with age until they forage as far as adults [25,64]. For example, *R. ferrumequinum* juveniles begin foraging near the roost and, by 50 days of age, match adult foraging distances, with weaning estimated at 45 days [25]. Food intake increases rapidly between days 29 and 55 [25], and foraging time and efficiency (higher intake in shorter foraging time) increase with age [89]. Similarly, even if *R. euryale* juveniles use the same habitats as adults [63], adults repeatedly exploit distant, specific hunting hot spots [90] that juveniles do not know yet, as locating distant prey patches requires learning and memorising resource locations [91,92,93].

The disparity of degrees of dissimilarity between juvenile and adult diets observed in this study indicates that the relative importance of psychomotor development, foraging strategies, prey discrimination, or spatial learning might differ among species—or localities. Along with our third hypothesis, the species with higher trophic specialisation—*R. euryale* and *R. hipposideros*—exhibit more significant differences in diet between juveniles and adults than the more generalist *R. ferrumequinum*. In this scenario, attaining a high level of specialisation may require a more extensive learning process.

Bat conservation critically depends on identifying essential trophic resources through diet studies [65,94,95,96]. However, most studies focus on species-level needs, overlooking age-specific requirements. Juvenile survival during the first winter is crucial for understanding bat population dynamics, with juveniles generally having lower survival rates than adults [97]. Seasonal changes in prey availability also impact bat diets [60,61,62]. Although further research is needed, these variations may influence dietary specialisation and alter the differences between juvenile and adult diets. Our results suggest that juvenile bats do not solely depend on the most profitable prey that adults select. Instead, at least in some development stages, they may also heavily rely on less profitable but easier-to-catch prey, which may be crucial for their early development and fat storage before their first winter.

## 5. Conclusions

In conclusion, our study provides novel insights into the dietary differences between adult and juvenile bats across three horseshoe bat species, with varying degrees of intraspecific diet specialisation. While juveniles consistently showed broader diets than adults, significant age-related dietary differences were observed in *Rhinolophus hipposideros* and *R. euryale*, but not in *R. ferrumequinum*. We found a shared pattern, where juveniles consume smaller and smoother prey that adults. These differences are likely shaped by factors such as prey availability, foraging skills, and psychomotor development. Furthermore, our results emphasise the role of learning in juvenile foraging behaviour, with specialised prey selection emerging as bats mature, especially in the lesser and Mediterranean horseshoe bats. The more generalist diet pattern of the greater horseshoe bat and the importance of individual specialisation can explain the lack of significant differences in this species. These findings contribute to a more nuanced understanding of bat ecology, with implications for conservation efforts, particularly in recognising the distinct dietary needs of juveniles for their survival and successful development.

## Figures and Tables

**Figure 1 animals-14-03303-f001:**
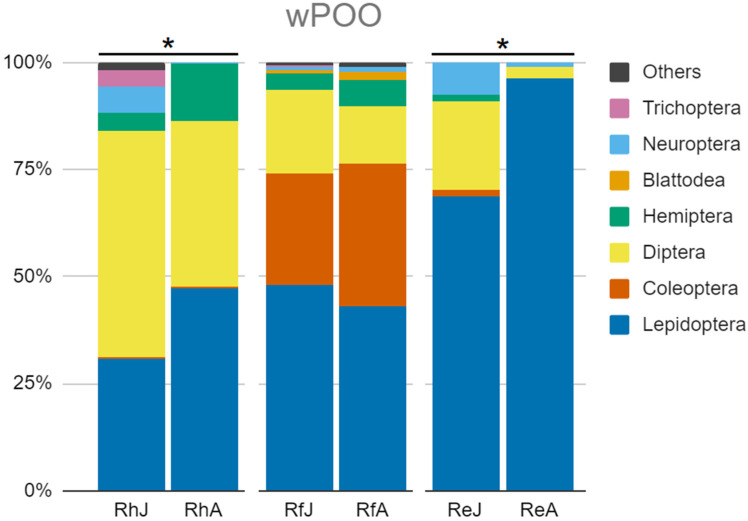
The diet composition of the six species–age classes of bats is expressed as “weighted percentages of occurrences” (wPOOs) [46] of their main prey merged at the order level. (RhJ: *R. hipposideros* juveniles; RhA: *R. hipposideros* adults; RfJ: *R. ferrumequinum* juveniles; RfA: *R. ferrumequinum* adults; ReJ: *R. euryale* juveniles; ReA: *R. euryale* adults). The asterisk points out significant differences between juvenile and adult diets showed by PERMANOVA analysis at the species level.

**Figure 2 animals-14-03303-f002:**
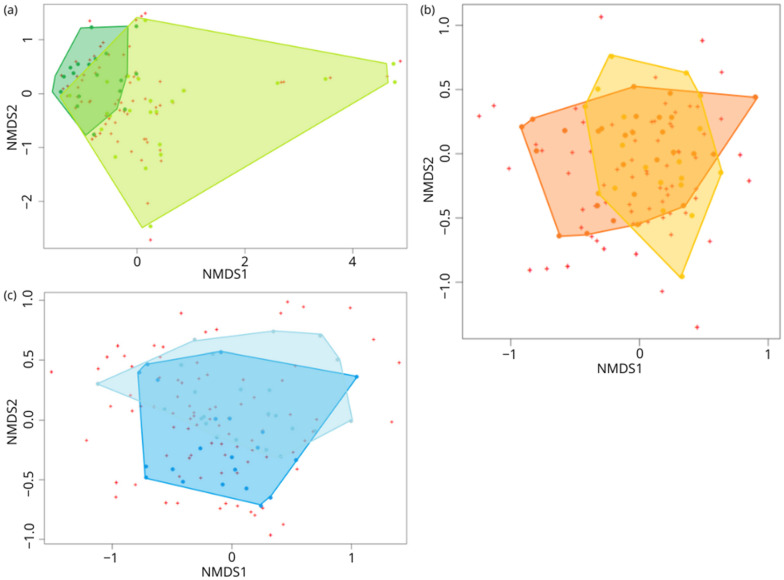
NMDS (non-metric multi-dimensional scaling) ordination of the bat individuals’ diets. (**a**) *R. euryale* (green), (**b**) *R. ferrumequinum* (orange)and (**c**) *R. hipposideros* (blue).Light and dark colours indicate juveniles and adults, respectively. Each dot in the graphs represents a bat individual.

**Figure 3 animals-14-03303-f003:**
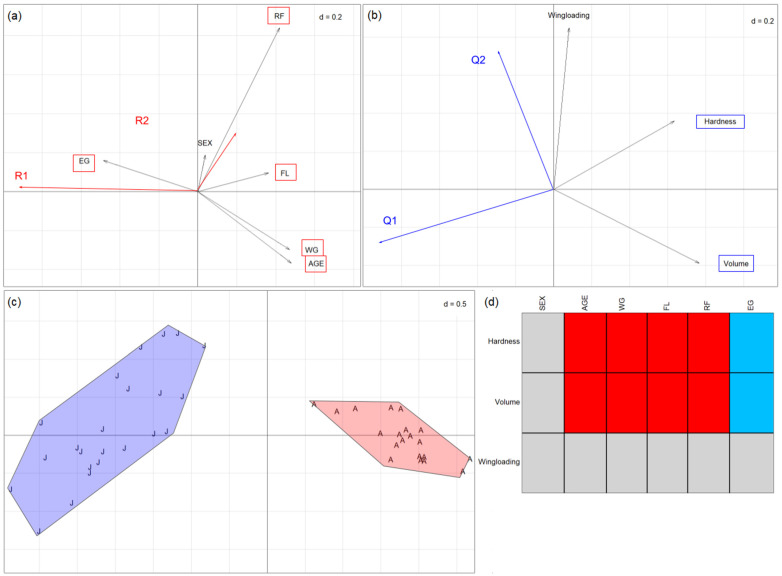
Results of RLQ analyses for the diet of *R. euryale* individuals: (**a**) coefficients for bat traits (EG: epiphyseal gap; FL: forearm length; RF: resting frequency; WG: weight), (**b**) coefficients for prey traits (traits framed in a rectangle represent a significant association with the axes), (**c**) eigenvalues and scores of bat individuals (red A: adults; blue J: juveniles), and (**d**) results of the 4th-corner analysis. Red cells represent positive associations, and blue cells represent negative ones (*p*-value < 0.05). Panels display the first two axes only, with d-values referring to grid size. Monte-Carlo test: observed statistic = 1.793 (standardised observed statistic = 5.678), with *p*-value < 0.0001.

**Figure 4 animals-14-03303-f004:**
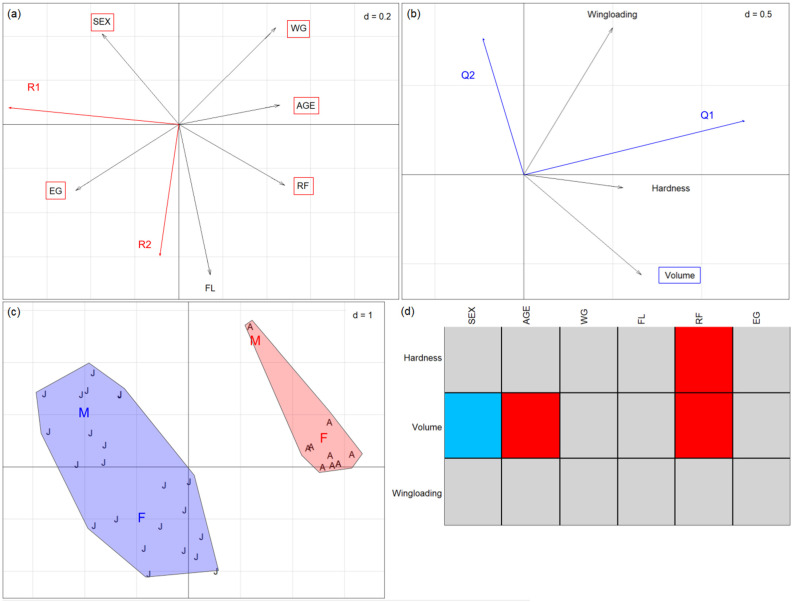
Results of RLQ analyses for the diet of *R. hipposideros* individuals: (**a**) coefficients for bat traits (EG: epiphyseal gap; FL: forearm length; RF: resting frequency; WG: weight), (**b**) coefficients for prey traits (traits framed in a rectangle with dotted lines represent an association with the axes (*p* < 0.05)), (**c**) eigenvalues and scores of bat individuals (red A: adults; blue J: juveniles), M represent the grouping of males and F of females, and (**d**) results of the 4th-corner analysis. Red cells represent positive associations, and blue cells represent negative ones (*p*-value < 0.05). Panels display the first two axes only, with d-values referring to grid size. Monte-Carlo test: observed statistic = 0.786 (standardised observed statistic = 6.3348), with *p*-value < 0.01.

**Figure 5 animals-14-03303-f005:**
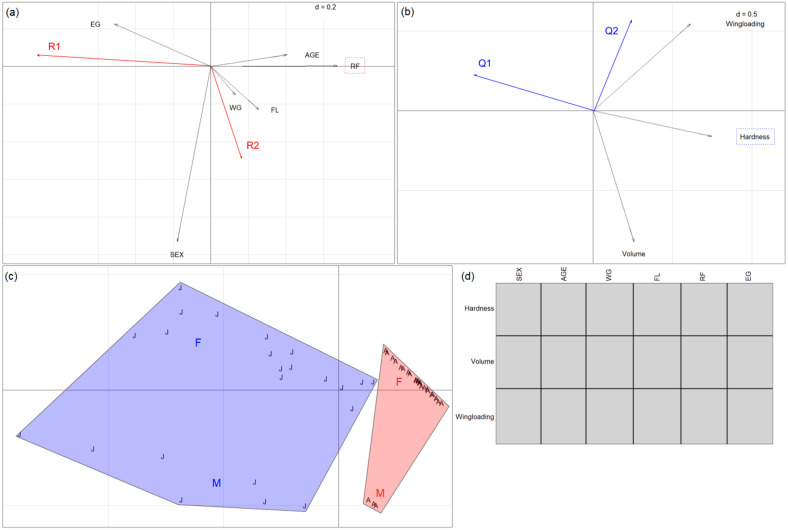
Results of RLQ analyses for the diet of *R. ferrumequinum* individuals: (**a**) coefficients for bat traits (EG: epiphyseal gap; FL: forearm length; RF: resting frequency; WG: weight), (**b**) coefficients for prey traits (traits framed with dotted lines in a rectangle represent an association with the axes (*p* < 0.1)), (**c**) eigenvalues and scores of bat individuals (red A: adults; blue J: juveniles), M represent the grouping of males and F of females, and (**d**) results of the 4th-corner analysis. Red cells represent positive associations, and blue cells represent negative ones (*p*-value < 0.05). Panels display the first two axes only, with d-values referring to grid size. Monte-Carlo test: observed statistic = 0.054 (standardised observed statistic = −0.175), with *p*-value 0.451.

## Data Availability

The original contributions presented in the study are included in the article/Appendix A; further inquiries can be directed to the corresponding author.

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
