# Peer review of "Learning to Hunt on the Go: Dietary Changes During Development of Rhinolophid Bats"

_animals, 2024, doi:10.3390/ani14223303_

Round 1
Reviewer 1 Report
Comments and Suggestions for Authors
This manuscript is a welcome addition to the field, addressing an often-overlooked aspect of bats' trophic ecology: how development stages affect diet composition. It is a hypothesis-driven, meticulously conducted, and very well-written study that investigates the dietary shifts between juvenile and adult bats of three horseshoe species (Rhinolophus euryale, R. hipposideros, and R. ferrumequinum) in the Basque Country. With up-to-date methods like metabarcoding and trait analysis, the authors uncover significant dietary differences that highlight the distinct needs and hunting adaptations of juvenile versus adult bats.
This is one of the first times in my career I am pleased to recommend acceptance with virtually no comments. My very few comments are as follows:
Some references that might be worth considering:
Regarding R. euryale foraging behaviour:
- Russo, D., Jones, G. and Migliozzi, A., 2002. Habitat selection by the Mediterranean horseshoe bat, Rhinolophus euryale (Chiroptera: Rhinolophidae) in a rural area of southern Italy and implications for conservation. Biological Conservation, 107(1), pp.71-81.
- Russo, D., Almenar, D., Aihartza, J., Goiti, U., Salsamendi, E. and Garin, I., 2005. Habitat selection in sympatric Rhinolophus mehelyi and R. euryale (Mammalia: Chiroptera). Journal of Zoology, 266(3), pp.327-332.
Regarding sexual segregation in vespertilionids:
- Russo, D., Jones, G., Polizzi, M., Meola, V. and Cistrone, L., 2024. Higher and bigger: How riparian bats react to climate change. Science of the Total Environment, 913, p.169733.
Also, some scientific names need italicization. Please double-check that when revising your text.
Author Response
We sincerely appreciate your time and effort in reviewing this manuscript. Below are our detailed responses and the corresponding revisions and corrections highlighted in the track changes document of the resubmitted files. I specified the location of the change in the revised manuscript (not the marked version). Thank you for your valuable feedback.
Please see the attachment

Reviewer 2 Report
Comments and Suggestions for Authors
Overall, I would like to congratulate the authors on their manuscript "Learning To Hunt On The Go: Dietary Changes During Development of Rhinolophid Bats". The study is clear, the underlying concept is novel and intriguing, and I enjoyed reading your paper. After evaluating it, I have included a few minor comments for your consideration in the attached file.

Author Response

(The authors gave the same response as above.)

Reviewer 3 Report
Comments and Suggestions for Authors
Overall, I think this was an interesting study on diet differences betwene adult and juvenile bats, and how prey traits may influence those differences. This is an important bit of natural history information for considering conservation of bat species. The methods are fairly straight forward and I feel that the conclusions are supported by the analyses done. Here are some more specific thoughts and suggestions that I think will improve the strength of the manuscript:
One overall note: inconsistent use and formatting of species scientific and common names. Additionally, I believe that the species names need to be italicized (though check with editors for confirmation).
Abstract:
Line 29: Mentions localities, but nothing else in abstract talks about comparing between locations, just between ages. Recommend just removing that word, as it is also not explicitly discussed anywhere else in the manuscript.
Introduction:
Line 41: I think dietary changes have been very well studied in bats, so this statements reads a little false. Do the authors mean specifically ontogenetic changes in diet? If so, then recommend adding that detail.
Line 48: Can you provide a citation about memorizing locations of distant food resources? While it seems plausible, it also feels a bit speculative.
Line 51: Previous sentence says “optimal foraging models” but then the sentence starting on line 51 says “According to this theory”. Theory and models are different things, so recommend clarifying exactly what is meant (models are based on the theory). Note: this comes up again in the Discussion, so just make sure the wording matches in both places.
Line 58: This paragraph feels like it would fit better if it followed the paragraph that first discussed echolocation (then diving into optimal foraging theory stuff).
Line 61: What is meant by “handling” the calls? Can you clarify on what you mean by “deep learning process”? This sentence is also very long and covers a lot, so might be better served being separated out into separate thoughts. For the example citation, also recommend a short explainer as to WHY it matters that juvenile echolocation resting frequencies are lower than adults, especially since there is a decent amount of time spent discussing frequencies in the discussion.
Line 80: naming three different groups of insects as potential prey suggests a decently varied diet for R. hipposideros. Do you just mean that they are more narrow in the scope of prey size?
Last paragraph/hypothesis: You mentioned the third hypothesis as being related to trophic specialization. However, none of the previous background discussed this type of prey detail (only size and taxonomic information). Can you add additional background to clarify what you mean by trophic specialization? From my perspective, all insect eaters are feeding on the same trophic level. Given that your methods don't really address this, I think you can remove this from the hypotheses (and maybe add something in the discussion about future directions).
Methods:
Noting the difference in sampling times for the different colonies (2021 for one and differnet parts of August 2022 for the other two). Can you provide any additional information in the discussion about why or why not that cuold affect your results. It gets touched upon, but it feels very hand-wavey - would love to see more specific citations.
Line 116: can you clarify on how resting calls were recorded? Bats in bags, in hands etc? Also, I am wondering: what is the relationship between resting frequency and the call frequencies that bats may be using when actively hunting.
Results:
Diet composition - The diets of hipposideros and ferrumequinium appear to be similarly balanced to me (jus switching Coleoptera for Hemiptera), so I’m not sure you can justify saying hipposideros is primarily made up of Lepidoptera with less than 50%.
Line 216 - instead of saying adonis analysis, just say the PERMANOVA. You already told us the specifics, so when referring back to it, I think the general analysis term is fine (and easier to understand than the specific function used). *same comment applies to the caption for Figure 1.
Line 226: This line reads kind of awkward, recommend making more active. It is also a bit confusing as someone who is not super familiar with Pianka’s index. Is it a measure of distance or is a measure of percent similarity? Might be helpful to have a quick sentence that provides some grounding in what is being measured (and be careful with wording thins like “distant niches”).
Line 243/244: What is “coinertia”?
Figure 3 - no in text discussion of 3c or 3d - I had to refer to the caption to understand what those panels were. I’m also not sure about Figure 3 - 5 a being separate. It might be more useful to see these different elements compared between species, instead of showing all of the analysis for one species at a time. The ordination graphs might not be necessary and could be moved to the supplement (just telling us verbally what the loadings are is fine in my opinion, as the visuals are hard to read and don’t really say anything you can’t just say in words).
For R. hipposideros - is the sexual dimorphism enough to warrant doing these analyses separately? I understand if the sample size is not there, but it would be helpful to know if the relationships seen apply to both sexes (even if just a quick supplement analysis). It also seems interesting to me that resting frequency and prey hardness are associated…I would anticipate that kind of things to be more affected by body size or something like jaw strength.
Discussion:
Line 340: remove the bit about aligning with initial hypothesis. It breaks up the sentence awkwardly. Instead, add another sentence saying how that conclusion did or did not meet your predictions.
Line 343: remind me what are the bat development indicators? or Instead, just say directly what trait you are talking about (body size, echolocation frequency etc).
Line 435: This first sentence doesn’t feel necessary and doesn’t relate specific to the rest of the paragraph.
Line 349: again, using the terms trophic resources - it is not clear what that means in the context of this study. Instead, just say prey species abundance/availability.
Line 351: is that true? It looked to me based on Figure 2 that juvenile ferrumequinum actually seem to show more restricted diet composition than adults.
Line 359 - 363: This paragraph feels out of place, and may be better placed towards the end of the discussion as it breaks up the discussion about diet in R. euryale and comparison to hte previous study.
Line 375 - differences observed don’t address the methodological issues. I think you mean to say that you feel your differences are robust to the potential sampling/methodology issues? Can you provide more detailed citations on why you feel that way?
Line 377 - sudden switch to bat common name after using scientific names for the rest of the manuscript.
Line 377 - Line 385 - this detailed discussion of previous paper should probably be grouped with other discussion of that same paper, especially since it focuses on just the one species studied.
Line 394: This relationship doesn’t make sense to me as worded. The “if-then” format of hte sentence implies that adults feeding on harder/larger prey than juveniles is also responsible for the difference between sexes. Also, were sex differences compared between all bats sampled, or only within age demography (male vs female adults, male vs female juveniles)?
General discussion point: A little table of the different morphometric and trait parameters measured on bats and insects might be helpful. The discussion talks about things like “slower prey”, but it is unclear what traits were measured that related to that, making it hard to follow.
Comments on the Quality of English LanguageFor the most part, I found the paper to be quite readable, though there are a few typos and missing punctuation scattered throughout.
Author Response

(The authors gave the same response as above.)

Reviewer 4 Report
Comments and Suggestions for Authors
The manuscript tested how the diets three horseshoe bat species differed betwen juveniles and adult, and between sexes by analysing DNA in bats feces. In addition to that, it also compared how prey used by each age categoyy differered based on some traits. In general the manuscript is well written and organized, and I only have some minor comments.
General comments: should species names be in italics along the text?
Line2 9-10. Please, fix this part of the sentence “By analysing bat the DNA 9 in their droppings,…”
Line 41-42. Please, provide a framework to undertand why this is relevant for general readers not familiar with bats.
Line 105: “00:30 hours”
Lines 173-174. Please fix “… diets. (93 out of the 119 species 173 constituting the core diets (Supporting Information 2) were …” Check the ().
Line 297 “..we found the biggest..”
Lines 317 and 218. With p values close to 0.1, you use “weak association” and “no significant difference”. It seems that in line 371 you are using the language of evidence for p, but not in line 218.
Line 351. Broader in all 3 spp or only in 2 species?
Line 359 Change to “Trait-based diet studies play a crucial role…..”
Lines 360-361 “Our dietary comparisons between adult and juvenile bats, contribute…”
Line 405 “this behavior”.
Line 438. “This means that…”
Line 450. “ and by 50 days of age match...”
Line 454 “..that juveniles do not know…”
Author Response

(The authors gave the same response as above.)
